

# Kinematic characterization of backhand stroke accuracy in squash based on kinematic variables between two different skill levels—a preliminary cross-sectional study

Rui Huang[*], Haojie Li[*], Jian Jiang, Zhitao Zhou and Chen Xiu

Shanghai University of Sport, Shanghai, China
[*] These authors contributed equally to this work.

## ABSTRACT

**Background**. The aim of this study was to compare differences in shot placement and accuracy between national and international level squash players. Squash is a technically demanding sport and understanding the biomechanical characteristics of athletes at different levels is important for developing effective training strategies.

**Methods**. The study used a three-dimensional motion analysis system, a high-speed video camera, and a professional tee for biomechanical testing. Participants included national and international level squash players. The kinematic characteristics of the shoulder, elbow, and wrist joints of the upper extremity for backhand strokes, as well as the accuracy of the strokes, were analyzed to compare the differences between the two groups of athletes in terms of stroke posture and accuracy.

**Results**. The kinematic analysis of the backward backhand stroke revealed that national squash players showed significant differences compared to international players in several key parameters. Specifically, national players had significantly greater trunk flexion ($P = 0.018$) and less shoulder medial rotation ($P = 0.027$). They also had lower racket velocity in the X-direction ($P = 0.043$). However, there were no significant differences in trunk lateral flexion ($P = 0.487$), trunk rotation ($P = 0.293$), shoulder extension/flexion ($P = 0.396$), elbow flexion/extension ($P = 0.818$), wrist flexion/extension ($P = 0.177$), wrist rotation ($P = 0.476$), racket pitch ($P = 0.112$), racket velocity in the Y-direction ($P = 0.587$), or racket velocity in the Z-direction ($P = 0.327$). Additionally, data for racket yaw, racket roll, and racket Vx were not provided with significant values, indicating that these parameters do not show significant differences.

**Conclusion**. International level players outperformed national level players in squash stroke accuracy and control. Key kinematic factors influencing accuracy include trunk forward flexion, shoulder abduction, shoulder internal rotation, and racket angles. The study recommends that coaches design training to enhance technical details and positional control to improve squash performance.

Corresponding author
Chen Xiu, xiuchen@sus.edu.cn

## INTRODUCTION

Squash is a competitive sport with varying techniques and tactics, attracting more than 15 million players from 135 countries worldwide (*Sinclair, Hobbs & Selfe, 2015*). The players' common knowledge about squash is that technique is the basis of successful tactical implementation (*Jones et al., 2018*). Different types of squash strokes can be used to hit the squash ball during competitions, considering the location of the ball relative to the court and the location of the opponent (*Vučković et al., 2014*). Successful performance in a squash match requires the ball to be hit accurately to strategically advantageous areas of the court, thereby rendering it difficult for an opponent to successfully return the ball to their advantage (*Jones et al., 2018*; *Vučković et al., 2014*; *Lees, 2003*). Thus, any technique is aimed toward scoring and winning.

Within the competitive squash community, a ball hit from very close to the side or back wall is commonly perceived to be more difficult than a hit that is close to the front wall (*Vukovi et al., 2003*). A previous study demonstrated that elite players are highly capable and prefer hitting the ball closer to the side or back wall more frequently than less elite players (*Jones et al., 2018*; *Vuckovic et al., 2005*; *Mitchell, 2006*). Additionally, the backhand drive rate was observed to be 59% for the back-wall drive skill (*Vuckovic et al., 2005*; *Hong, 1996*). Therefore, backward backhand drive may be considered an essential factor in the frequency of use, score ratio, and match outcome.

A fundamental aspect of sports biomechanical support for athletic groups is conducting assessments of differences in technical performances at different skill levels for identifying the key biomechanical factors affecting technical performance. Several studies have investigated the biomechanical characteristics of backhand during squash matches. *Kawamura et al. (2017)* studied the kinematic characteristics (*i.e.,* spent time and displacement) of joints and rackets during backhand drop shots. *Vuckovic et al. (2005)* analysed game time and distance covered in squash competitions using two-dimensional video analysis. *YouSin (2007)* examined the differences between the kinematics and muscle activities depending on the changes in the angle of the approaching balls in the backhand drive during squash. Although previous studies have provided some technical analysis of squash, we are still unclear about the differences in backhand kinematics and stroke accuracy among squash players at different levels. Therefore, there is an urgent need for us to conduct this study.

Previous studies have shown that the backhand stroke in squash shares kinematic similarities with frisbee throwing, emphasizing the pivotal roles of trunk, upper arm, forearm, and hand movements in achieving high-velocity strokes (*Lees, 2003*; *Goktepe, 2009*). Understanding these segmental kinematics across different skill levels is crucial for developing precise and powerful strokes essential for success in squash matches (*Williams et al., 2020a*). Moreover, ball placement accuracy is a fundamental skill in racket sports, strongly influencing a player's tournament ranking (*Kawamura et al., 2017*). Therefore, investigating the specific kinematic factors contributing to variations in ball placement accuracy among squash players of varying skill levels can offer strategic insights to enhance competitive advantage and increase points won during matches.

This study aimed to systematically examine the differences in joint kinematics and ball location accuracy in the squash backhand stroke between international and national squash players. These assessments of biomechanical differences would provide useful information for coaches, practitioners, and athletes. This study focuses on the technical characteristics of high-level squash players, aiming to analyze the differences in the techniques of high-level squash players, therefore, this study provides important scientific advice for competitive squash events, in addition, although this study is for high-level kinesiology, it also has a positive significance for the general applicability of squash because for the majority of squash beginners, the ability to carry out fast and scientific is very important, and scientific technical training has a positive effect on the prevention of sports injuries, as well as the ability to better and faster master the correct and reasonable techniques, so by analyzing the techniques of high-level athletes, it is beneficial to guide the training of beginners. This study also has wide applicability. In summary, this study provides important guidance for establishing scientific training methods to improve the technical performance of squash players. It can also have a positive impact on the popularization and promotion of squash worldwide.

## MATERIALS AND METHODS

### Study design

A biomechanical cross-sectional comparison study was conducted to identify the differences in the joint kinematics and ball location accuracy in the squash backhand stroke among the international and national squash players.

### Subjects

Eleven male squash players volunteered to participate in this study. Each participant was allocated to either the national ($n = 6$) or international group ($n = 5$). The domestic group consisted of the best squash players in China, selected from the Chinese national squash team and top three athletes at nationals (age $22.50 \pm 4.11$ years, height $1.85 \pm 0.04$ cm, weight $74.72 \pm 7.41$ kg). Participants in the International Division are selected from the top 50 players in the World Professional Squash Association rankings, according to the National Teams athletes and who finished in the top three in the world competition (age $27.60 \pm 2.80$ years, height $1.79 \pm 0.06$ cm, and weight $72.48 \pm 7.66$ kg). All the participants-related details are provided in Table 1. All methods in this study were carried out following the 'Declaration of Helsinki' relevant guidelines and regulations, and this study was reviewed and approved by the Ethics Committee of the China National Rehabilitation Center (No.S20220203). Permission was obtained from all participants with written informed consent.

### Equipment setup

All biomechanical tests were performed at an indoor squash training facility at the Shanghai University of Sport. The study experiment system comprised a 3D motion analysis system, high-speed video cameras, and a professional serve machine. (1) A 12-camera 3D motion analysis system (Qualisys Track Manager 2.9, Medical AB, Sweden) was used to collect
**Table 1 Basic characteristics of study participants.**

| Subject | Group | Age (year) | Height (m) | Weight (kg) | Rank (PSA) |
|---------|-------|------------|------------|-------------|------------|
| 1 | National | 22 | 1.90 | 70.10 | None |
| 2 | National | 25 | 1.79 | 68.50 | None |
| 3 | National | 18 | 1.89 | 69.30 | None |
| 4 | National | 24 | 1.82 | 75.50 | None |
| 5 | National | 17 | 1.88 | 74.70 | None |
| 6 | National | 29 | 1.83 | 90.20 | None |
| 7 | International | 29 | 1.84 | 84.30 | 21 |
| 8 | International | 26 | 1.82 | 76.80 | 37 |
| 9 | International | 29 | 1.80 | 72.40 | 32 |
| 10 | International | 23 | 1.68 | 62.70 | 26 |
| 11 | International | 31 | 1.79 | 66.20 | 45 |

Notes.
B, badminton training experience; S, squash training experience; PSA, Professional Squash Association.

the marker trajectory data at a sampling frequency of 200 Hz; (2) two high-speed video cameras (Phantom LC111, Vision Research Inc., USA) were used to record the squash impact ground location in the target area at a sampling frequency of 50 Hz; and (3) a professional serve machine (TDinga, Ding-ge, China) was placed 2.5 m away from the front wall. This equipment was used to simulate the drive ball at a launching angle of 40° and a ball speed of 140 km/h, as played by professional athletes in real games. A sketch of the ball trajectory is shown in Fig. 1. Twelve infrared cameras were placed on the four corners of the court and synchronized by synchronizers connected at the same height; two cameras were placed on the two sides of the court (Fig. 1). In addition, global and local coordinate systems were defined for the target area, as shown in Fig. 1.

## Testing protocol

In this study, the testing protocol was based on the Hunt Squash Accuracy Test (HSAT), which requires the players to hit defined target areas on a squash court. This has been shown to exhibit strong correlations between tournament shot success and the percentage of winning shots during a match (*Williams et al., 2018*). The target areas (50 cm × 50 cm) were marked on the court with masking tape, as shown in Fig. 1.

Participants performed self-selected warm-up exercises that included warming the ball to be "match ready". Prior to the start of data collection, the participants wore loose-fitting shorts and shoes. Fifty-one 14 mm infrared-reflective markers were fastened to each participant's body, according to the Vicon plug-in-gait model marker set. Three reflective markers were attached to the mid-sides and top of the racket head.

In the normal testing, the participant stood behind the service box line and was instructed to move swiftly and hit the drive ball with a backhand stroke after the ball from the professional serve machine bounced off the floor. The ball was required to bounce off the front wall and fall into the target area. The participants were instructed to hit 50 drive shots continuously and were requested to perform each shot at a speed similar to that in a

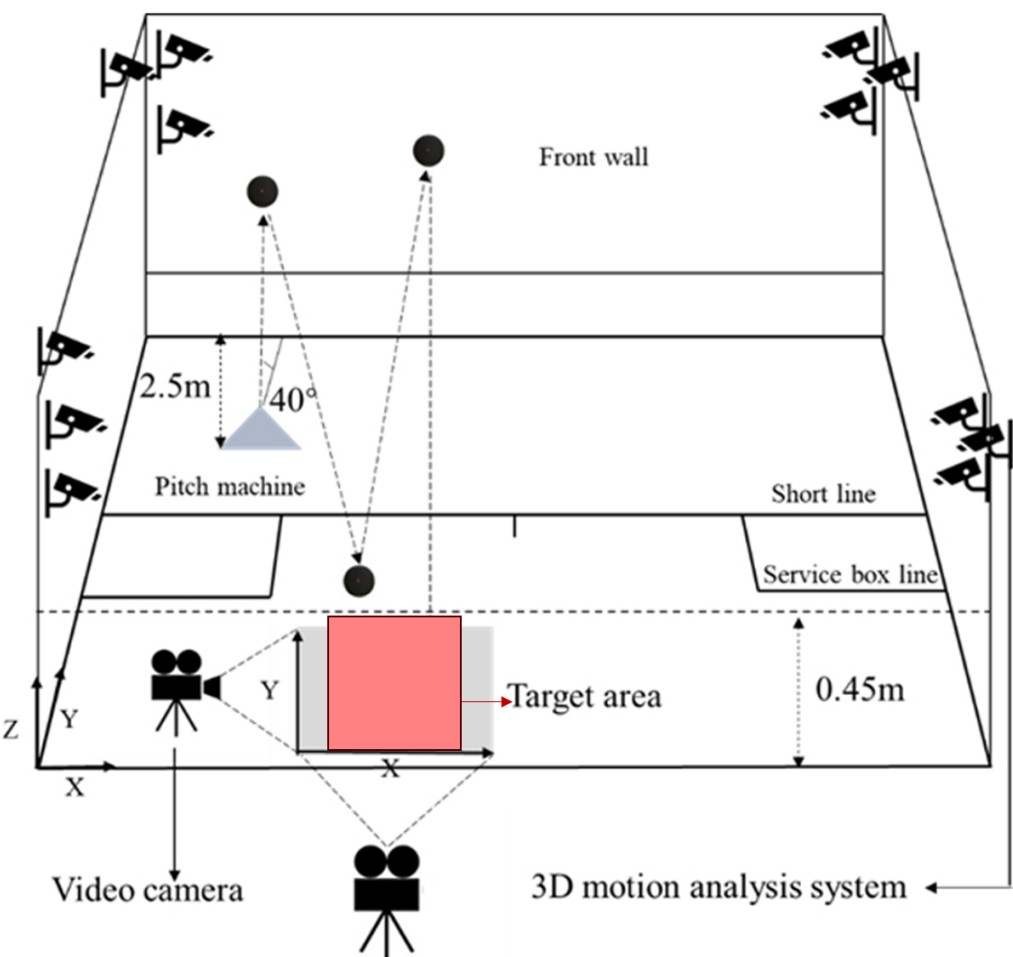

**Figure 1** Equipment setup: the target area marked on the floor of the court, camera positions around the squash court, and global coordinate system.

real game. Each test was conducted under standard environmental conditions at the same time of the day.

## Data analysis

The preliminary processing of kinematics was performed using Qualisys Track Manager to delete redundant markers, and the intercept phase was then exported as C3D files. The 3D kinematics were processed using the Visual 3D (v3, C-Motion Inc., United States) software, after which the marker trajectory data were smoothed using a low-pass Butterworth 4th order at a cut-off frequency of 6 Hz (*Decker et al., 2003*). From the kinematic data, a 15-segment biomechanical model (head, trunk, pelvis, 2 forearms, 2 upper arms,

Two thighs, two shanks, and two feet were created using Visual 3D. The squash swing was divided into two phases: downswing and follow through. The event variables were analysed based on three periods: top-swing (the time at which the marker on the racket

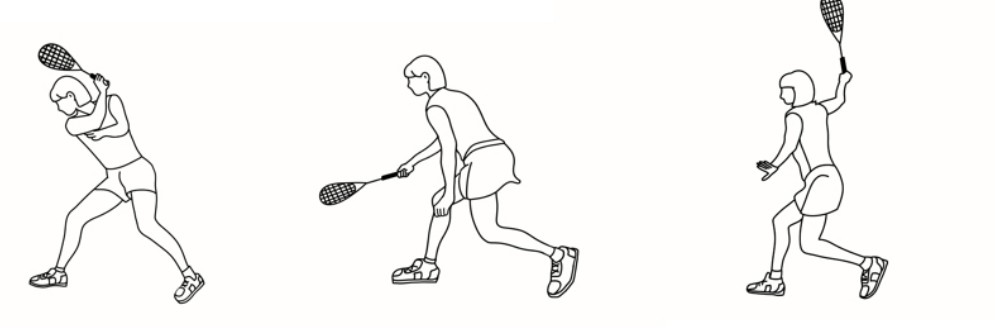

1.Moment of swing    2.The moment the ball touches the racket    3.Swing end moment

**Figure 2  Backhand stroke technique motion diagram.**

attains the peak vertical height), impact (the time at which the squash is in contact with the racket), and finish (*Kim, Min & Subramaniyam, 2018*).

The squash backhand stroke technical movement is divided into three moments, being the start of the swing—the touch phase—the end of the swing, this a complete backhand stroke (shown in Fig. 2).

The importance of racquet kinematics in sport has been validated in several studies (*Kwon, 2017*; *Williams et al., 2024*). The present study builds on this foundation and further explores the key role of the racket in sport performance. Based on the definitions of previous studies (*Williams et al., 2024*), we explicitly defined a local reference coordinate system for the racket, which included specific directions for the $y$-axis (yaw), $x$-axis (pitch), and $z$-axis (roll). $y$-axis: perpendicular to the racket face, used to describe the direction of rotation of the racket in the horizontal plane. $x$-axis: perpendicular to the long axis of the racket, parallel to the racket face, used to describe the rotation or tilt of the racket in the vertical direction. z axis: parallel to the long axis of the racket, used to describe the rotation of the racket around its own axis (Fig. 3). This definition provided a clear theoretical framework for the subsequent study, enabling us to systematically analyze the racquet's action performance under different motion states. During the experimental process, we accurately calculated the pitch, yaw, and roll angles at the moment of contact between the ball and the racket in order to establish the precise spatial localization of the racket with respect to the reference coordinate system in the laboratory. The measurement of these angles not only contributes to the kinematics, but also provides experimental data to support a deeper understanding of the racquet movement process. By analyzing these data, we can further explore how the racquet affects the flight path, reaction speed, and overall smoothness of the movement in sports performance.

In addition, the joint kinematics of the upper limbs and trunk can significantly affect a player's performance. This has become a common perspective among international squash coaches and in this research field (*Williams et al., 2020b*; *Woo & Chapman, 1992*). Therefore, we calculated the special value at impact and the 3D joint angle curve of the trunk

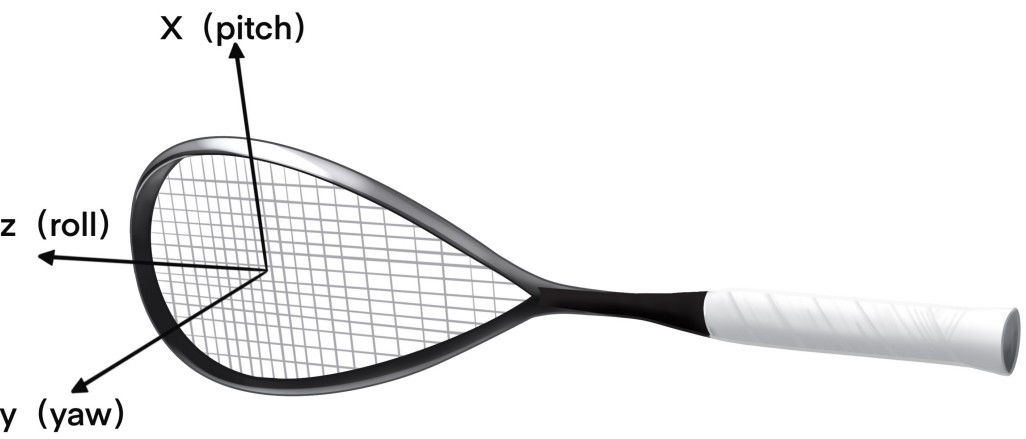

**Figure 3** Schematic diagram of the tennis racket coordinate system.

and upper limb during the squash backhand. All kinematic waveforms were normalised to 100% of the movement. Kinematic data are categorized into positive and negative values, with positive values being joint flexion and adduction and negative values being joint extension and abduction (*Li et al., 2023a*).

*Kawamura et al. (2017)* proposed the 95% confidence ellipse to be an effective indicator for assessing pitch location accuracy. Therefore, two parameters were measured in the present study to assess squash location accuracy: the percentage of location in the target area and the location accuracy, which is evaluated by considering the dimensions of the location distribution area using the 95% confidence ellipse method.

## Statistical analysis

All statistical analyses were performed using IBM SPSS (version 20.0; SPSS Inc., Chicago IL., USA). All calculated variables for national and international players were first subjected to the Shapiro–Wilk (SW) test. The SW test is used to test whether the test is derived, the joint kinematics, racket angle, racket velocity, and 95% confidence ellipse area did not show a normal distribution. The test is derived, the joint kinematics, racket angle, racket velocity, and 95% confidence ellipse area did not show a normal distribution. The Mann–Whitney U test was applied to test the group differences. It is used to compare whether the medians of two independent samples are significantly different. Other continuous variables showing normal distribution were tested using one-way ANOVA. We used the chi-square test (chi-square test) to assess the ball drop accuracy ratio. The chi-square test is a statistical method used to compare the difference between observed and expected frequencies, and is particularly useful for analyzing associations between categorical variables. In this case, upper extremity joint kinematics are compared using the maximum value for comparison, which in turn analyzes the change in the maximum joint range of motion when hitting the ball with the backhand, and stroke accuracy is compared using the mean value. Maximum values were not used for stroke accuracy due to the need to normalize by eliminating differences in subjects' single stroke accuracy. The significance level was set at $p < 0.05$.

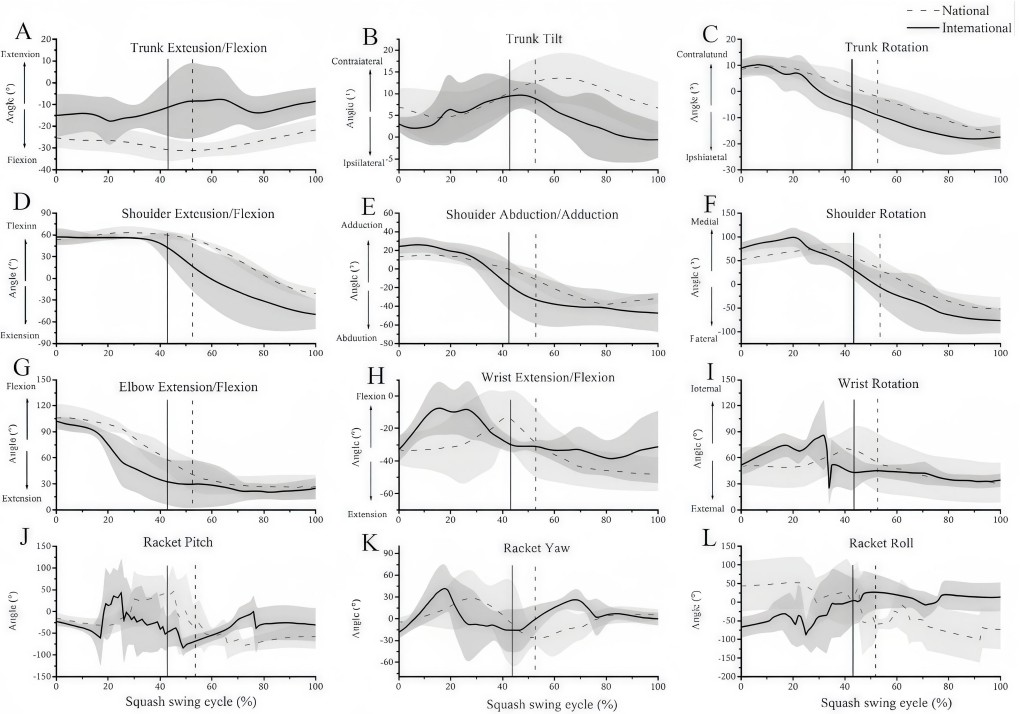

**Figure 4 Kinematics of joints and racket in backhand between two groups.** (A) Trunk extension/flexion, (B) trunk mediolateral (ML) tilt, (C) trunk rotation, (D) shoulder extension/flexion, (E) shoulder abduction/adduction, (F) shoulder rotation, (G) elbow extension/flexion, (H) wrist extension/flexion, (I) wrist rotation, (J) racket pitch, (K) racket yaw, and (L) racket roll. The impact time in the national group (52.24% ± 9.44%) and the international group (41.80% ± 15.48%) are plotted as black dashed and solid vertical lines, respectively. All kinematic waveforms were normalised to 100% of the movement. Kinematic data are categorized into positive and negative values, with positive values being joint flexion and adduction and negative values being joint extension and abduction.

## RESULTS

### Kinematics of joints and racket-related in backward backhand between two groups

Figure 2 shows the kinematics of the trunk (Figs. 4A–4C), shoulder ((Figs. 4D–4F), elbow (Fig. 4G), wrist (Figs. 4H and 4I), and racket (Figs. 4J–4L) in the backhand between the two groups. During impact, the shoulder kinematics of national players in the frontal plane was observed to be abduction, whereas that of international players was adduction (Table 2). Compared to the international group, the national group showed a higher trunk flexion angle, lower shoulder medial rotation angle, and smaller racket velocity in the X-direction during impact ($P < 0.05$, Table 2). In addition, the yaw and roll of the racket were significantly different between the two groups during impact (Table 2). However, no significant differences were observed between the two groups in the other kinematic parameters ($P > 0.05$, Table 2).

**Table 2   Kinematics of joints and racket at impact moment between two groups (mean ± SD).**

| Parameters | National group | International group | F | P |
|---|---|---|---|---|
| Trunk Flex (−) / Ext (+) °* | −32.64 ± 4.10 | −22.06 ± 4.88 | 9.337 | 0.018 |
| Trunk lateral flexion Contra (+) / Ipsi (-) ° | 13.27 ± 6.01 | 10.32 ± 4.63 | 0.537 | 0.487 |
| Trunk rotation Contra (+) / Ipsi (-) ° | −2.13 ± 2.47 | −5.15 ± 4.12 | 1.295 | 0.293 |
| Shoulder Ext (-) / Flex (+) ° | 54.16 ± 7.60 | 58.09 ± 3.55 | 0.816 | 0.396 |
| Shoulder Abd (-) / Add (+) °# | −9.54 ± 5.65 | 1.28 ± 5.69 | – | – |
| Shoulder rotation M (+) / L (-) °* | 33.10 ± 8.17 | 49.76 ± 7.65 | 7.717 | 0.027 |
| Elbow Ext (+) / Flex (-) ° | 32.08 ± 10.57 | 30.41 ± 7.90 | 0.057 | 0.818 |
| Wrist Ext (-) / Flex (+) ° | −26.57 ± 8.06 | −16.58 ± 9.25 | 2.258 | 0.177 |
| Wrist rotation Internal (+) ° | 57.34 ± 19.33 | 65.63 ± 8.88 | 0.566 | 0.476 |
| Racket pitch° | 57.57 ± 34.88 | 1.65 ± 44.32 | 3.312 | 0.112 |
| Racket yaw°# | −36.12 ± 61.37 | 73.25 ± 3.98 | – | – |
| Racket roll°# | −18.59 ± 86.14 | 0.70 ± 77.63 | – | – |
| Racket Vx (m/s)* | 1.31 ± 0.62 | 3.73 ± 1.64 | 6.061 | 0.043 |
| Racket Vy (m/s) | 15.43 ± 1.51 | 16.24 ± 2.12 | 0.324 | 0.587 |
| Racket Vz (m/s) | 7.23 ± 10.71 | 1.47 ± 1.11 | 1.109 | 0.327 |

Notes.

Flex, flexion; Ext, extension; Contra, contralateral; Ipsi, ipsilateral; M, medial; L, lateral.

Confidence intervals were greater than 0.95, with $P$-values representing significant differences and $P < 0.05$ indicating variability.

F-values were used to compare the degree of variability between between-group differences and within-group errors; * Reflects a significant difference between the two groups; # Reflects the movement pattern was different between the two groups and without statistical analysis.

### The squash location accuracy in backhand between two groups

Figure 5 shows the squash locations with a 95% confidence ellipse for a typical player from the national (Fig. 5A) and international (Fig. 5A) groups with 50 drives. The squash location accuracy was significantly lower in the national group (45.57% ± 10.37%) than in the international group during the backhand stoke (84.29% ± 10.91%, $F = 46.316$, $P < 0.01$, Fig. 5B). Furthermore, the 95% confidence ellipse area of squash locations of the international group (2197.14 ± 210.07 cm$^2$), was lower than that of the national group (2449.71 ± 167.35 cm$^2$, $F = 6.19$, $P = 0.029$, Fig. 5B).

## DISCUSSION

In this study, we compared the joint kinematics and ball accuracy of the squash backhand stroke between international and national players. Results showed significant differences in trunk flexion, shoulder abduction, shoulder rotation, and racket kinematics between the groups. International players also demonstrated better ball location control. These findings highlight the link between biomechanical differences in backhand technique and ball control, offering valuable insights for improving squash training.

Our findings showed that national players had greater trunk forward flexion, less shoulder internal rotation, and lower racquet speed in the $X$-axis during backhand strokes compared to international players. National players likely use trunk flexion for stroke control, while international players emphasize shoulder coordination and racquet speed. Differences in racquet yaw and roll also suggest varying control over ball spin and rebound

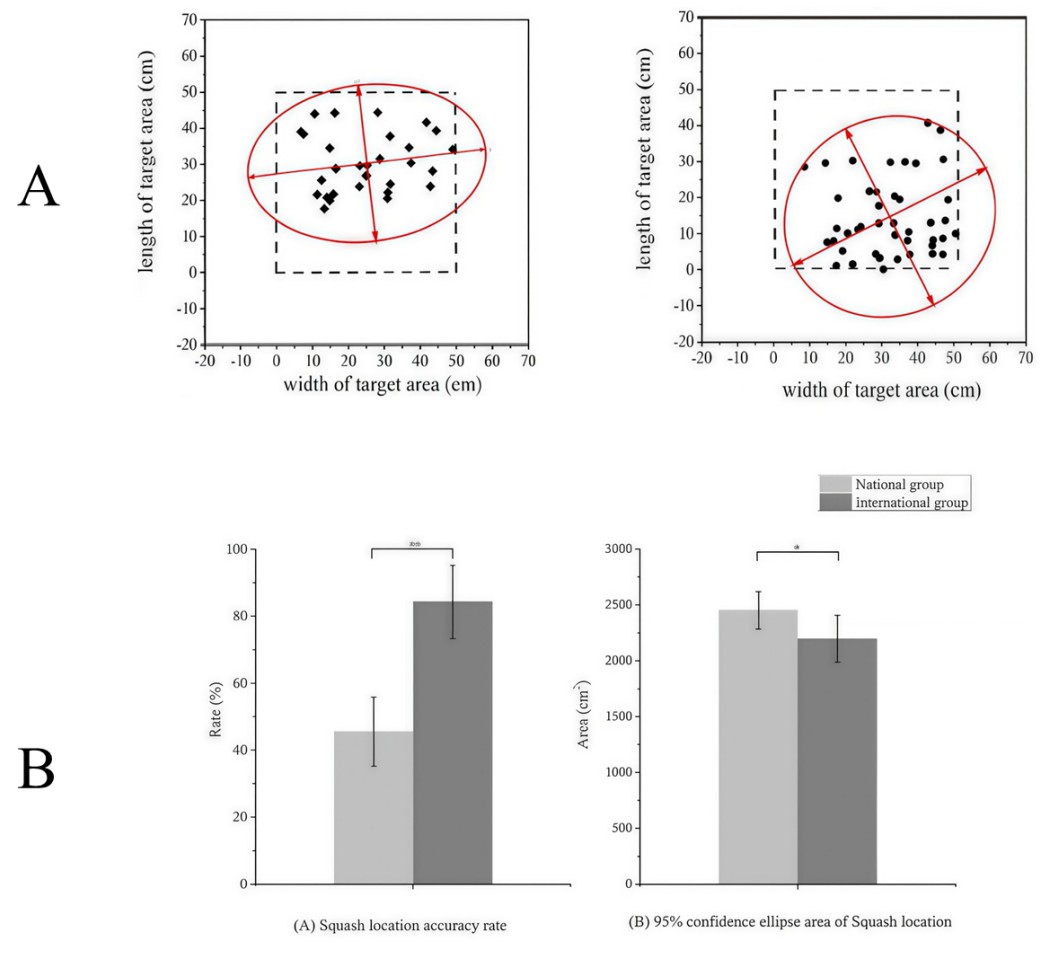

**Figure 5** **The squash location with 95% confidence ellipse for one typical player in each group with 50 drives.** Comparison of the (A) squash accuracy and (B) 95% confidence ellipse area for the backhand stroke between the national and international groups.

path, impacting stroke accuracy and difficulty. National players may focus on accuracy and control, while international players enhance ball speed and spin, affecting strategic choices and match performance. Previous studies link squash location accuracy with tournament rankings (*Kawamura et al., 2017*; *Williams et al., 2020b*; *Shinya et al., 2017*). More importantly, it was proposed that the number of points lost owing to unforced errors in squash matches could be dependent on the skill level (*Mitchell, 2006*). In this study, we compared backhand stroke accuracy between international and national squash players using two-dimensional video analysis. Results revealed that national players had lower accuracy and larger 95% confidence ellipses compared to international players. National players' shots were concentrated in the front of the target area, while international players' were focused on the back and sides. These findings underscore the link between location accuracy and performance, though direct comparisons with previous studies are limited. Racket orientation during impact is identified as a crucial factor for accuracy in racket sports (*Davey, Thorpe & Williams, 2002*; *Sheppard & Li, 2007*; *Williams et al.,*

*2020b*). *Williams et al. (2020a)* reported that inaccurate backhand drop shots exhibited a significantly smaller racket roll angle (racket faceless open) at impact than accurate shots, indicating that this parameter was a determining factor in the accuracy of this stroke. In this study, the yaw and roll angles of the racket differed significantly between national and international players during impact, with angles in opposite directions. Although the national group had a larger racket pitch angle ($57.57 \pm 34.88$) compared to the international group ($1.65 \pm 44.32$), the difference was not significant. These variations likely contribute to the observed differences in squash location accuracy. Coaches should develop targeted training programs to address these discrepancies and help players maintain stable hitting patterns during high-intensity competitions.

Our study found that national-level players had significantly lower backhand stroke accuracy compared to international players, reflecting challenges in technical execution and stroke control. This may be due to national players' kinematic characteristics, such as higher torso flexion and lower racket speeds, which affect ball control. International players demonstrated better accuracy and a more concentrated hitting zone, indicating better control and consistency. They also likely generate more scoring opportunities and adapt more effectively to opponents' tactics. National players could improve their performance by focusing on technical training to enhance backhand accuracy and control. Additionally, racket velocity at impact has been noted as a crucial performance factor in racket sports (*Williams et al., 2019*). A highly positive correlation between racket velocity and players' performance was reported in previous studies (*Jones et al., 2018*; *Kim, Min & Subramaniyam, 2018*). *Kim, Min & Subramaniyam (2018)* demonstrated that a novice group maintained a faster racket velocity than an expert group, which reflects the inexperience of novice groups. However, no significant difference was observed in the racket velocity in the anterior–posterior direction in the current study. The reason for the lack of difference in racket velocity may be related to the experimental design of this study; that is, the players were required to hit the ball on the target area. The "Accuracy First" strategy may be selected when the participant satisfies the experimental requirements and sacrifices speed for accuracy (*Li et al., 2023b*).

The hit is generated by the addition of actively coordinated segments that transfer the power generated from the ground into a well-developed kinematic chain, resulting in the final acceleration of the racket onto the ball (*Goktepe, 2009*). Therefore, small variations in the segment kinematics can cause a change in the racket-related paraments, resulting in low performance (*Hughes & Franks, 1994*). A previous study suggested that longitudinal rotations about the trunk and upper arm segments, shoulder adduction, and wrist flexion are segmental movements that primarily contribute to racquet motion during impact (*Vuckovic et al., 2005*; *Kibler et al., 1996*; *Elliott, Takahashi & Noffal, 1997*). In the current study, the shoulder kinematics of national players in the frontal plane was abduction, whereas that of international players was adduction during impact. In addition, compared to the international group, the national group showed a higher trunk flexion angle and a lower shoulder medial rotation angle during impact. *Elliott, Takahashi & Noffal (1997)* showed the rotation of the upper arm to be important, because it is one of the primary sources of racket speed for the backhand stroke. Therefore, international players

hitting the ball with shoulder adduction and medial rotation may exhibit certain advantages, including increasing the work distance in the follow-swing phase and trajectory control of the racket. Interestingly, the increase in trunk flexion in the national group may reflect the poor rationality of technical applications, as it reduces the efficiency of power transmission. Furthermore, the poor controllability of the lower limb results in the flexing of the trunk when receiving and serving a low ball. These kinematic differences suggest that the coaches should focus on correcting the technical details, by increasing the corresponding training for solidifying or strengthening the player's technical control.

This study has several limitations. The small sample size and observed differences in technical movements among elite athletes highlight the need for larger studies to explore biomechanical differences between player levels. While we compared kinematics and squash location accuracy, we did not examine their correlation. Future research should include electromyography and kinetic analysis to better understand the biomechanical factors affecting technique and accuracy, guiding training strategies. Expanding the sample size and improving study design, such as allowing free ball hits and measuring additional variables like muscle activity and joint range of motion, could offer a more comprehensive view of squash technical characteristics.

## CONCLUSIONS

International players exhibited a higher squash location accuracy and controllability than national players. Trunk flexion, shoulder abduction, shoulder medial rotation, and racket-related angle may be the primary biomechanical factors affecting squash accuracy control and performance. This study recommends coaches to design corresponding training that strengthens the technical details and squash location control, which is meaningful for improving squash performance.

### Funding
The authors received no funding for this work.

### Competing Interests
The authors declare there are no competing interests.

### Author Contributions
- Rui Huang conceived and designed the experiments, performed the experiments, analyzed the data, prepared figures and/or tables, and approved the final draft.
- Haojie Li performed the experiments, prepared figures and/or tables, and approved the final draft.
- Jian Jiang conceived and designed the experiments, prepared figures and/or tables, and approved the final draft.
- Zhitao Zhou performed the experiments, analyzed the data, authored or reviewed drafts of the article, and approved the final draft.

- Chen Xiu analyzed the data, authored or reviewed drafts of the article, and approved the final draft.

## Human Ethics

The following information was supplied relating to ethical approvals (*i.e.*, approving body and any reference numbers):

and this study was reviewed and approved by the Ethics Committee of the China National Rehabilitation Center (No.S20220203)

## Data Availability

The raw data is available in the Supplementary File.

## Supplemental Information

Supplemental information for this article can be found online at http://dx.doi.org/10.7717/peerj.18333#supplemental-information.

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
