# Peer review of "Kinematic characterization of backhand stroke accuracy in squash based on kinematic variables between two different skill levels—a preliminary cross-sectional study"

_PeerJ, doi:10.7717/peerj.18333_

## Round 0.1 · original submission · Minor Revisions

The authors should provide a careful review of the manuscript, answering each reviewer point-by-point and making the needed changes. Please, provide more details on methodological procedures and data analysis, as requested by reviewers.

Reviewer 1 ·

Basic reporting

1. Why not directly title it "Kinematic Characteristics"? Wouldn't that be more precise?

2. It would be better to provide specific numerical results in the abstract.

3. The conclusion in the abstract is too general. Specific conclusions based on the research results should be provided.

Experimental design

1. In the abstract, "…by analyzing joint motion and stroke phases.", "trunk" is not usually referred to as "joint. “and "joint motion" and "stroke phases" are not parallel relationships.

2. The participants in this study are of a very high level, but are the differences found universally applicable? How do you address the issue of individual differences? Additionally, in line 71, the study purpose mentions, "Furthermore, it can positively impact the popularity and promotion of squash globally." How does this study positively impact the global popularity and promotion of squash?

Validity of the findings

1. Were the participants in the national group selected from the national team? It is recommended to provide more detailed grouping criteria.

2. The article mentions that most of the participants in the national group were originally badminton players, but it does not provide a detailed analysis of how their training background may have affected the research results. It is suggested to increase the analysis and discuss the potential impact of the participants' training background on the research results.

3. The "target area" in the figures does not appear to be a square, and the resolution of other images is too low to be clear.
4. The results section mentions differences in certain kinematic parameters between the two groups, but does not delve into the specific impact of these differences on actual game performance. It is suggested to discuss the potential implications of these differences in actual game performance.

5. What do the solid and dashed lines perpendicular to the X-axis in Figure 2 represent? Is there any statistical analysis of sequential data differences during impact?

6. It is recommended to label the annotations directly on the numerical values for clarity.

7. Figure 3 and Figure 4 could be combined.

8. The discussion section mentions some limitations of the study but does not provide detailed suggestions for future research directions. It is recommended to add detailed recommendations for future research directions, such as how to improve research design and what variables to measure, to provide guidance for subsequent research.

Reviewer 2 ·

Basic reporting

The study offers valuable insights into the biomechanical differences between squash players of varying skill levels. With minor revisions for clarity and additional details, it will significantly contribute to sports science. Please refer section 4 for the comments.

Experimental design

Please refer section 4 for the comments.

Validity of the findings

Please refer section 4 for the comments.

Additional comments

1. In the results section of the abstract, could you please include a comparison of the joint kinematic metrics between national and international level squash players?
2. It appears that the sentence on line 54 is incomplete. Could you please revise it for clarity?
3. Including a figure of a player performing a backhand stroke in an actual arena would greatly enhance the paper.
4. Could you please provide a figure illustrating the local coordinate system mentioned in lines 138-141?
5. For Figure 2, an illustrative figure of a dummy hand with axes showing the positive and negative directions of each kinematic measurement would help readers better understand the measurement system.
6. It would be helpful to clarify whether the statistical evaluation compared only the maximum/mean values or if multiple time instances were compared. Please provide more details.
7. For Table 2, please consider adding confidence intervals to the data presented.
8. The discussion effectively links the findings to the research question. However, a deeper analysis of the practical applications of these findings for coaches and athletes would be beneficial.
9. Please elaborate on how future studies might incorporate electromyography and kinetic analyses.
10. Merging parts A and B of Figure 3 into a single figure would improve clarity and facilitate better comparison.
11. The manuscript should confirm that all identifiable information, such as player ranks, has been removed from the data.
12. Providing the raw or processed data would facilitate future use by other researchers.

·

Basic reporting

1. Good manuscript writing is done in English.
2. The introduction needs a detailed exploration of selected variables. I suggest you improve the description on lines 57-65 to provide more detailed information about variables.

Experimental design

Line no 102 “terminal box of an analogue/digital converter. The 12 infrared cameras were positioned in the” be specific where its, was “analogue or converter” used in your study.

Validity of the findings

1. The authors had taken care of all mandatory requirements of good experimental research work by obtaining ethical approval and having written consent from participants.
2. The discussion made by authors in this manuscript was very well supported by studies related to findings.
3. Complete raw data is provided in the manuscript.

Additional comments

1. Chi-square test. (Analysis missing which is mentioned in line no 160.) .
2. Complete raw data is provided in the manuscript which is food practice. But in table no 1, Badminton training experience for each national level player is given whereas none of the International players had badminton training experience. It seems something is missing please recheck it.

---

## Round 0.2 · Minor Revisions

Congratulations on the work. However, there are some issues presented by the reviewers that should be carefully answer/clarified. Please, provide the needed changes and answer to reviewer, and specifically regarding results. These changes and improvements are essential.

Reviewer 1 ·

Basic reporting

abstract
1.The Results section presents some inconsistencies in the reporting of statistical values, as certain findings include F-values while others do not. It is recommended to maintain consistency throughout by either including or omitting F-values uniformly.
2.Furthermore, rather than using the general expression "P < 0.05," the specific P-values reported in the original data should be presented to enhance precision and clarity.
3.The section could specify which other kinematic parameters showed no significant differences (P > 0.05) to provide a fuller understanding of the study.
4.Please simplify the language in the Results and Discussion sections of the abstract to make them more concise.

Experimental design

no comment

Validity of the findings

no comment

Reviewer 2 ·

Basic reporting

Significant improvements and corrections were made based on my previous feedback. The tracked changes were clear and addressed the concerns effectively clearly.

Experimental design

-NA-

Validity of the findings

-NA-

Additional comments

-NA-

·

Basic reporting

No Comment

Experimental design

No Comment

Validity of the findings

No Comment

Additional comments

No Comment

---

## Round 0.3 · accepted · Accept

The authors have addressed all the reviewers comments and this manuscript is ready for publication. Some minor changes can be asked during the following stages.